# A Convolutional Neural Network with Fletcher–Reeves Algorithm for Hyperspectral Image Classification

**Chen Chen [1], Yi Ma [1,2,\*] and Guangbo Ren [2]**

[1]  College of Geomatics, Shandong University of Science and Technology, Qingdao 266590, China;
   chenchencc@fio.org.cn
[2]  First Institute of Oceanography, Ministry of Natural Resources, Qingdao 266061, China;
   renguangbo@fio.org.cn
\*  Correspondence: mayimail@fio.org.cn; Tel.: +86-532-88967094

**Abstract:** Deep learning models, especially the convolutional neural networks (CNNs), are very active in hyperspectral remote sensing image classification. In order to better apply the CNN model to hyperspectral classification, we propose a CNN model based on Fletcher–Reeves algorithm (F–R CNN), which uses the Fletcher–Reeves (F–R) algorithm for gradient updating to optimize the convergence performance of the model in classification. In view of the fact that there are fewer optional training samples in practical applications, we further propose a method of increasing the number of samples by adding a certain degree of perturbed samples, which can also test the anti-interference ability of classification methods. Furthermore, we analyze the anti-interference and convergence performance of the proposed model in terms of different training sample data sets, different batch training sample numbers and iteration time. In this paper, we describe the experimental process in detail and comprehensively evaluate the proposed model based on the classification of CHRIS hyperspectral imagery covering coastal wetlands, and further evaluate it on a commonly used hyperspectral image benchmark dataset. The experimental results show that the accuracy of the two models after increasing training samples and adjusting the number of batch training samples is improved. When the number of batch training samples is continuously increased to 350, the classification accuracy of the proposed method can still be maintained above 80.7%, which is 2.9% higher than the traditional one. And its time consumption is less than that of the traditional one while ensuring classification accuracy. It can be concluded that the proposed method has anti-interference ability and outperforms the traditional CNN in terms of batch computing adaptability and convergence speed.

**Keywords:** convolutional neural network (CNN); Fletcher–Reeves algorithm (F–R); conjugate gradient; coastal wetland classification; hyperspectral imagery

## 1. Introduction

Remote sensing image classification is important for environmental monitoring. The emergence of hyperspectral imaging technology and hyperspectral remote sensing imagery provides more possibilities for remote sensing image classification. The information richness of hyperspectral imagery also challenges the traditional image classification methods.

Deep learning [1] spread rapidly to various fields at an alarming rate, and made major breakthroughs in many fields such as speech recognition [2,3], image classification [4,5], text understanding [6], etc. Additionally, some methods were developed for the classification of hyperspectral remote sensing images, such as the methods based on sparse representation [7], metric learning [8], stacked autoencoder

(SAE) [9–11], stacked sparse autoencoder (SSAE) [12], etc., showing the effectiveness of classification. Deep learning is a multi-layer perceptron with multiple hidden layers [13]. Compared with the support vector machine (SVM) [14–16], boosting [17,18], maximum entropy method [19], etc. the shallow machine learning model with a hidden layer node or no hidden layer nodes, the deep nonlinear network structure of deep learning can learn the deep connection of data and effectively improve the classification accuracy. Deep learning, which occupied an important position in image classification, also shows great potential in remote sensing classification. As one of the deep learning expert models, convolutional neural network (CNN) proved its successful application in remote sensing image classification [20–24]. Many scholars improved the CNN model and applied it to hyperspectral imagery classification [25–28]. The general idea is to extract the spectral, spatial, or spectral–spatial jointly from hyperspectral images into the CNN model by dimension reduction [29], filtering [30], optimal band combination [31], etc., or enter the above resulting depth features to different classifiers such as logistic regression (LR) [29,32], extreme learning machine (ELM) [31] or SVM classifiers [30]. There are other excellent proposed feature extraction methods for CNN such as feature reconstruction [31], depth pixel-pair features extraction [33], semantic context-aware information extraction [34], hierarchical deep spatial extraction [35], spectral–spatial extraction with combined regularization [28] and so on. Due to the spectral and spatial richness of hyperspectral, many experts carried out related research on 3D CNN [36,37], and proposed the 3D CNN model in hyperspectral image classification, such as end-to-end spectral–spatial residual network (SSRN) [38], superpixel-based 3D deep neural networks [39], etc. In addition to improvements in network input, output and frame, there are a number of studies for the internal network optimization algorithm [40–42]. Most of the optimization algorithms are based on the gradient descent method for backpropagation to achieve a multi-layer network parameter adjustment [43,44]. Gradient descent has the advantages of simple, small calculation amount per iteration and less storage capacity in the process of optimizing parameters. However, the closer to the target value, the smaller the step size of the gradient descent method has and makes the progress slower. In addition to gradient descent-based optimization methods, there are also other proposed gradient update methods, such as Momentum [45], Nesterov Accelerated Gradient [46,47], etc., but they need to add additional parameters, which add complexity to some extent. As one of the classic optimization methods, the conjugate gradient algorithm can improve the search direction, that is, to construct a better search direction by linear combination of the past gradient and the gradient of the current point. Therefore, the conjugate gradient algorithm improves model convergence speed without adding any additional parameters. It not only has the advantages of a small calculation amount and storage capacity, but also improves the convergence speed [48], so it has unique advantages in large-scale engineering calculation.

In this paper, a CNN model based on the Fletcher–Reeves Algorithm [49] (F–R CNN) is proposed to solve the problem of slow convergence of the CNN model based on gradient descent algorithm. A method of increasing the number of samples by adding a certain degree of perturbed samples is further proposed to solve fewer optional training samples in practical applications. Experiments were carried out in changing the number of training samples and the number of batch training samples. The conclusion is drawn by comparing with the classification results of the traditional CNN model, and evaluating the proposed model in terms of anti-interference ability, batch computing adaptability and convergence speed.

The remainder of this article is arranged as follows. In Section 2, we introduce the data and the experiment area. The F–R CNN model is described in detail in Section 3. Then, we provide the results of the experiments and the discussion about this proposed work in Section 4. Finally, the conclusions are provided in Section 5.

## 2. Materials and Research Area

In this paper, the CHRIS (compact high resolution imaging spectrometer) hyperspectral remote sensing data covering the coastal wetland of the Yellow River Mouth National Nature Reserve is used

as the experimental data. The 0° image of the CHRIS working mode 2 of the study area in June 2012 is selected and it comprises 512 × 512 pixels.

## 2.1. Hyperspectral Data

CHRIS is mounted on the new generation of microsatellite PROBA (Project for On-Board Autonomy) launched by ESA (European Space Agency). It is a sun-synchronous orbit with a height of 615 km and a dip of 97.89°. As an imaging device, CHRIS consists of a telescope and a 770-column 576-line CCD sequence detection system imaging spectrometer, which acquires visible-near-infrared spectral data by push-sweep. The CHRIS sensor has five imaging modes: Mode 1 for land and water imaging, mode 2 for water imaging, modes 4, 5, and 6 for terrestrial imaging, and each can acquire five angles at the same location (0°, +36°, −36°, +55°, −55°) imaging. The specific parameters are shown in Table 1 [50].

**Table 1.** Compact high resolution imaging spectrometer (CHRIS) product description.

| Parameter | Index |
| --- | --- |
| Spatial sampling interval /m | End of the sky 18 |
| Image area/km × km | 14 × 14 |
| Number of images | 5(different angles) |
| Image size/km | 13 × 13(768 × 748pixels) |
| Each image size/Mbit | 131 |
| Pixel format | BSQ |
| Spectral range /nm | 400–1050 |
| Data unit | MicroWatts/nm/m2/str |
| Number of spectral bands | 18 bands with a spatial resolution of 17 m, 62 bands with 34 m |
| Spectral resolution | 1.3 nm@410 nm to 12 nm@1050 nm |
| Signal to noise ratio | 200 |

We chose the 0° image of the CHRIS working mode 2 as the study area. The spectral range of the image is 406–1035 nm, the spectral resolution is 1.25–11.00 nm, the number of bands is 18, and the ground resolution is 17 m. It has a wide spectrum and good imaging quality, which conforms to the imaging geometry of general hyperspectral remote sensing images, and is more suitable for the classification of ground objects in this study area. The HDFclean plug-in (Program for processing images stored in Hierarchical Data Format) provided by the ESA website is used to perform strip elimination and geometric correction on the CHRIS hyperspectral remote sensing image. The geometric correction error is sub-pixel, which satisfies the research requirements.

## 2.2. Supplementary Data

In order to accurately evaluate the classification results of the study area (shown in Figure 1a), it is necessary to produce a ground truth interpretation image of the data set. In 2012, two field surveys were conducted on the Yellow River Mouth wetland research area. According to the field data collected by the field survey and the high spatial resolution remote sensing image ZY-3 (spatial resolution 2.1 m), the ground type mark is established, as shown in Table 2.

Through the human-computer interaction interpretation, the ground type information extraction of the hyperspectral data was performed according to the type of interpretation mark in Table 2. Based on high spatial resolution image data and expert interpretation experience which is supported by the field work materials of the author's research group for many years, the interpretation result map was checked and corrected to generate the final ground interpretation result of the study area, as shown in Figure 1b. The ZY-3 high spatial resolution remote sensing image is shown in Figure 1c.

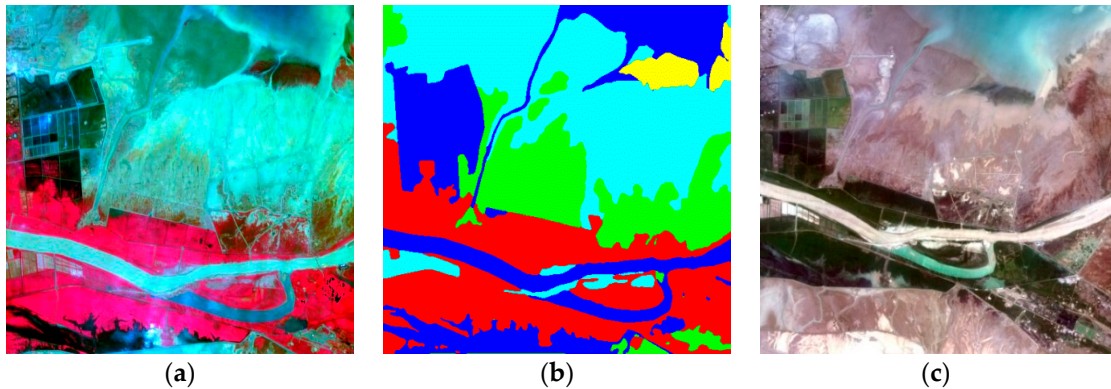

**Figure 1.** Study area data, ground truth and interpreting reference data. (**a**) False color image composite (band 15, 10 and 5); (**b**) ground truth image; (**c**) supplementary data (ZY-3 remote sensing image).

**Table 2.** Land cover types interpretation mark of the Yellow River Mouth wetland. Pattern represents CHRIS band 15, 10 and 5 false color composite images.

| No. | Color | Type | Pattern | Feature |
|---|---|---|---|---|
| 1 | | Reed | | Hi-wet, tall grass that can be wet for many years, can grow in fresh or salt water. |
| 2 | | Mixture of Suaeda-salsa and tamarisk | | The halophyte, which is high in humidity and salt-tolerant, grows on sand or sandy loam in coastal areas such as seashores and wastelands, and is sparsely distributed. Tamarix is a salt-tolerant shrub that grows on sand, silt and tidal flats such as beaches, beachheads, coasts, and is resistant to dry and watery, wind and alkali resistant. |
| 3 | | Water | | The river section near the Yellow River mouth. |
| | | | | Naturally formed or manually excavated pit water surface for aquaculture. |
| 4 | | Spartina alterniflora | | Plants resistant to salt and flooding, born in intertidal zone. |
| 5 | | Tidal flats and bare land | | This paper refers to the muddy tidal flat, the tide intrusion zone between the high tide level and the low tide level. |

## 2.3. Study Area Introduction

The Yellow River Delta wetland, located on the coast of the Bohai Sea in the northeastern part of Shandong Province, China, is one of the rare estuarine wetland ecosystems in the world. The wetland is located at the mouth of the Yellow River. Its geographical coordinates are 37°35′~38°12′N and 118°33′~119°20′E, with a total area of 153,000 hectares. The Yellow River Delta National Nature Reserve was established in 1992. There are rich wetland types and diverse landscape types, and the natural wetlands account for about 68.4% of the total wetland area. The vegetation coverage in the reserve is as high as 53.7%, forming the largest beach vegetation on the coast of China. The wetland ecosystem is a transitional stage from aquatic ecosystem to terrestrial ecosystem. Its basic ecological function is to regulate water cycle and maintain the flora and fauna resources of the wetland. Its vulnerability determines the reality of easy loss and difficult recovery. Therefore, research on the distribution of land cover types including vegetation in coastal wetlands is of great significance to protect its resources and environment. The study area is shown in Figure 2.

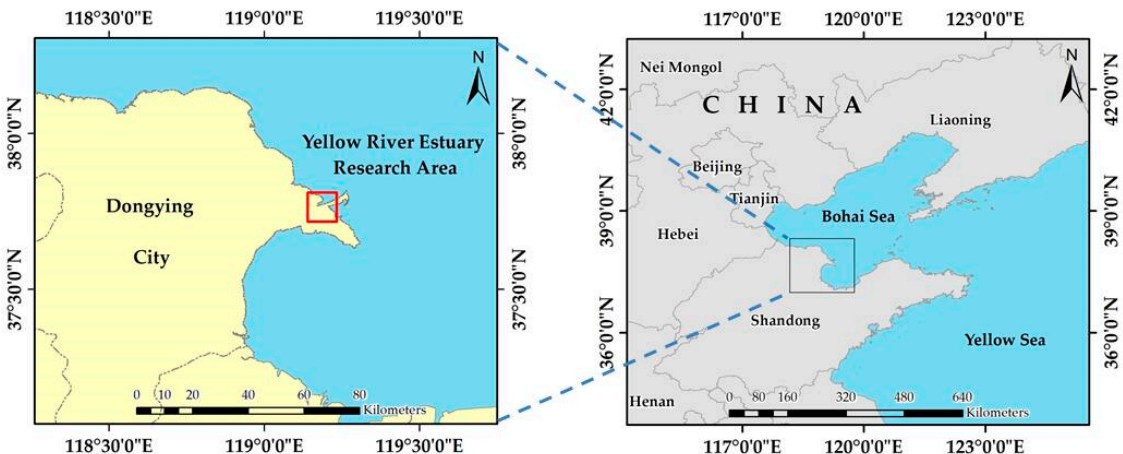

**Figure 2.** Location of the Yellow River Mouth wetland study area. The study area is enclosed by the red frame.

## 3. Proposed Method

In this section, we will detail the various parts of the proposed F–R CNN model. Figure 3 shows the overall framework of the proposed F–R CNN model. The framework mainly consists of two parts: Extraction of spectral features and the CNN model based on the F–R algorithm. For the input processing of hyperspectral imagery, the image spectral features are first normalized, and then converted into the form of input data. The F–R CNN classification module includes forward propagation for training network parameters, back propagation error for network parameter adjustment, and classification output. In the backpropagation process, we use the F–R algorithm, a classical method in the conjugate gradient method, instead of the gradient descent algorithm of the traditional CNN model to update the network parameters (weights and offsets). The extraction spectrum process of this work has a dimension-increasing process that doubles the spectral 18-dimensional replication to 36 in order to subsequent convolution operations. The parameters of the F–R CNN structure are shown in Table 3.

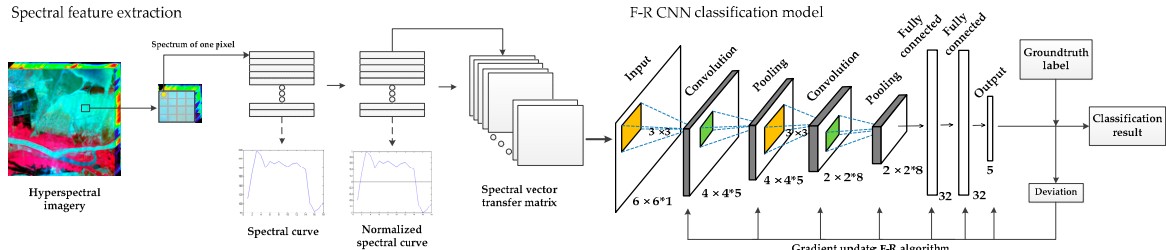

**Figure 3.** Proposed F–R CNN classification model. The structure includes the extraction process of spectral features and the convolutional neural network (CNN) classification model based on the Fletcher–Reeves (F–R) algorithm.

**Table 3.** The structure and parameters of the proposed F–R CNN.

| Layers | Layer Name | Image Size | Size of Kernels | No. of Kernels | Pool Scale |
|--------|-----------|-----------|-----------------|----------------|-----------|
| 1 | Input | 6 * 6 | - | - | - |
| 2 | Convolution | 4 * 4 * 5 | 3*3 | 5 | - |
| 3 | Pooling | 4 * 4 * 5 | 1*1 | - | 1 |
| 4 | Convolution | 2 * 2 * 8 | 3*3 | 8 | - |
| 5 | Pooling | 2 * 2 * 8 | 1*1 | - | 1 |
| 6 | Fully connected | 32 * 1 | - | - | - |
| 7 | Fully connected | 32 * 1 | - | - | - |
| 8 | Output | 5 * 1 | - | - | - |

### 3.1. Input Processing

Before inputting into the training model, the spectral features are processed. The first is to extract spectral vectors. Three-dimensional data with i-rows, j-columns and n-bands (i × j × n) is converted into two-dimensional information of m-samples and n-bands (m × n, m = i × j). The second is to normalize the extracted samples. We normalize the gray values of the preprocessed hyperspectral image sample data (including training samples and overall samples) between −1 and 1. Processing matrices by mapping row minimum and maximum values to [−1, 1]. Each row here represents the spectral vector of each sample. The algorithm formula is as follows.

$$y = (y_{max} - y_{min}) * \frac{x - x_{min}}{x_{max} - x_{min}} + y_{min},\tag{1}$$

where $y$ represents the normalized vector, $y_{max}$ and $y_{min}$ respectively represent normalized thresholds 1 and −1, $x$ is the sample spectral vector, and $x_{max}$ and $x_{min}$ are the maximum and minimum values of the sample spectral vector. The normalization of the spectral vector is a general method in machine learning problems that can help optimization methods (such as the gradient descent method of traditional models) converge faster and more accurately. If the sample gray value is directly used for classification, the result will be affected by the characteristics of the large variance, and the phenomenon that large numbers eat decimals between different scales will affect the classification accuracy. The third is to construct input data. In order to facilitate network input and convolution operation, the sample spectral vector data (m × n) is converted to the form of input data (x × y × m). In this experiment, we convert a matrix of 6090 × 36 into a matrix of 6 × 6 × 6090, where 6090 is the number of samples and 36 is the spectral dimension. The last is to create sample tags. Label the sample in the form of a unit vector according to the number of the land cover types in the table. For example, reed is the first type, its label is (1,0,0,0,0), and the mixture of Suaeda salsa and tamarisk is the second type, and its label is (0,1,0,0,0), and so on.

In the actual application process, due to the difficulty of the field data surveying and the small coverage of the research area, the training samples have certain limitations in the selection process. The small number of training samples would make the model over-fitting during training, because the feature learning is not comprehensive; and the large number of training samples would make the overall classification accuracy of the study area not objective enough and lose its applicability. Therefore, we thought of an expansion method with a small number of training samples based on the addition of perturbations to the samples. The basic idea is to combine the original training samples and the training samples with certain perturbations into new training samples as the input data of the model. The specific operation of generating a disturbed sample is to add a range of random values to the sample. For example, the disturbance range is [−0.01, 0.01] (the disturbance level is 1%), and the disturbance sample is generated by adding the random number generated in the range of [−0.01, 0.01] to the original sample. It is also possible to generate a disturbing sample data of two times or more of the original sample. The operation of the above-extended samples is performed after normalization. Figure 4 shows the process of generating a 2x training samples dataset with a disturbance level of 1%.

### 3.2. Mathematical Knowledge

In essence, the construction of the deep learning model needs the optimization method to optimize the objective function (or loss function) and train the best model. Common optimization methods include gradient descent, Newton's method, Quasi-Newton methods, conjugate gradient, etc. These methods are all calculated based on the derivative of the objective function.

The optimization idea of the gradient descent is to use the current position negative gradient direction as the search direction, because this direction is the fastest descent direction of the current position, so it is also called "the steepest descent method". The iterative formula for the steepest descent method [51] is:

$$x^{(k+1)} = x^{(k)} + \lambda_k \, d^{(k)},\tag{2}$$

where $d^{(k)}$ is the search direction starting from $x^{(k)}$, where the direction of the steepest descent at point $x^{(k)}$ is taken (Equation (3)). $\lambda_k$ is the step size for one-dimensional search starting from $x^{(k)}$ along the direction $d^{(k)}$. In the mathematical optimization method, the $\lambda_k$ is adjusted according to the optimization objective after each iteration, while in CNN, in order to reduce the amount of calculation, it is set to a fixed value, which is the learning rate that we are well known ($\alpha$ in Equations (8) and (9)).

$$d^{(k)} = -\nabla f\left(x^{(k)}\right), \tag{3}$$

The conjugate gradient algorithm is a classical optimization algorithm, and its basic idea is to combine the conjugation and the gradient descent algorithm to construct a set of conjugate directions as the direction of one-dimensional search by using the gradient at the known points [51]. The search direction in the conjugate gradient algorithm is shown in Equation (4):

$$d^{(k)} = -g_k + \beta_{k-1} d^{(k-1)}, \tag{4}$$

$$g_k = \nabla f\left(x^{(k)}\right), \tag{5}$$

$$\beta_i = \frac{\left\|g_{i+1}\right\|^2}{\left\|g_i\right\|^2}, \tag{6}$$

Wherein, $\beta_{k-1} = 0$ when $k = 1$; when $k > 1$, $\beta_{k-1}$ is calculated according to Equation (6), which is also a specific form of the calculation factor of the Fletcher–Reeves algorithm (F–R algorithm).

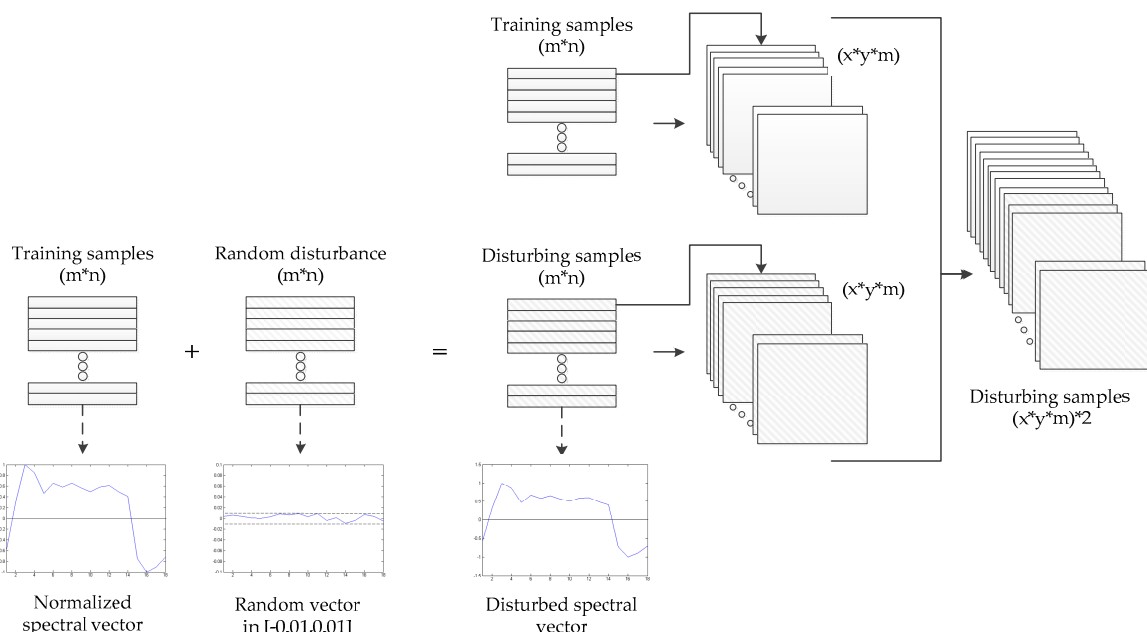

**Figure 4.** Increase the training sample. The disturbance sample data is generated by adding a random disturbance to the original sample. It shows an example of generating a data of 2x training samples. (The number of training samples: m = 6090, the spectral dimension: n = 36, and x = y = 6.).

The search direction of per iteration in the gradient descent algorithm is always perpendicular to the previous search direction. After multiple iterations, the sawtooth phenomenon [51] (Figure 5) tends to occur, which will cause the convergence speed to be slower near the minimum point. Newton's method, a method for approximate solving equations in real and complex domains, is characterized by its fast convergence. The conjugate gradient method is a method between the steepest descent and Newton's method, which only needs to use the first derivative information. It not only overcomes the shortcomings of slow convergence of the steepest descent, but also avoids the disadvantages of

Newton's method storage requirements and computational complexity of the Hesse matrix and its inversion. The figure below is a schematic diagram of the path comparison between the conjugate gradient method and the steepest descent method for searching for the optimal solution.

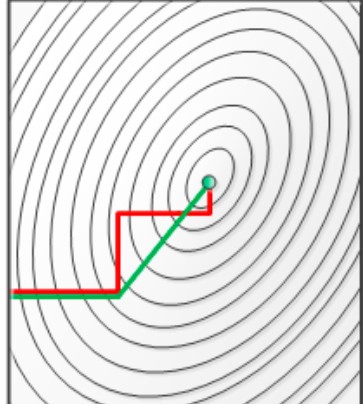

**Figure 5.** Comparison in searching for the optimal solution path. Green line indicates a conjugate gradient and red indicates a gradient descent.

In deep learning, a variety of gradient descent methods have been developed based on the gradient descent, such as stochastic gradient descent (SGD) [52], batch gradient descent (BGD) [53] and mini-batch gradient descent (MBGD) [54]. The difference between them is that each training session takes one training sample, all training samples and a certain amount of training samples for training the CNN model. Each method has its own advantages in terms of convergence speed or model optimization, but it cannot be balanced in most cases. In addition to this, some optimization algorithms need to be calculated with external parameters, such as Momentum [44], Nesterov Accelerated Gradient (NAG) [46] and so on. The conjugate gradient method is one of the most important optimization algorithms. It has the advantages of small-required storage, step convergence, high stability, and no external parameters compared to other optimization methods. The conjugate gradient method is not only one of the most useful methods for solving large linear equations, but also has advantages in solving large-scale nonlinear optimization problems.

### 3.3. F–R CNN Model

A complete CNN structure includes the input layer, convolution layer, pooling layer, fully connected layer and output layer. The input layer and output layer of this paper correspond to the spectral features and the category marks of land cover types. The basic structure is shown in Figure 3. The calculate process from the input layer to the output layer is called forward propagation. It calculates the cost function of the classification result and the ground truth types, and hopes that it is as small as possible, which makes the classifier's predictions as close as possible to the real land cover types. So we need to use an optimized method to adjust the network training parameters. This process is calculated in the opposite direction to the forward process, which is also known as backpropagation. Instead of using the gradient descent algorithm, we used the F–R algorithm in the F–R CNN model to update the parameters in the network.

The output obtained by a forward propagation has an error with the actual result, and the cost function $J(W, b; x, y)$ is used to calculate it:

$$J(W, b; x, y)_{batch} = \frac{1}{2k} \sum_{i=1}^{k} (h_{W,b,i}x - y_i)^2, \tag{7}$$

where $h_{W,b}(x)$ represents the predicted output, the subscript batch refers to the small batch samples, $k$ is the number of batch training samples, $x$ represents the input samples, and $y$ represents the actual

land cover types. The weight $W$ and the offset $b$ are parameter values trained by the F–R CNN network. In order to make the cost function $J(W, b; x, y)$ as small as possible, the parameter values of each layer are adjusted according to the error in the process of backpropagation until the cost function converges.

We use the F–R Algorithm to optimize the cost function in the F–R CNN model and the formula is as follows:

$$W_{ij}^{(l)} = W_{ij}^{(l)} + \alpha \Delta W_{ij}^{(l)}, \tag{8}$$

$$b_i^{(l)} = b_i^{(l)} + \alpha \Delta b_i^{(l)}, \tag{9}$$

In the above iterative formula of weights and offsets, $\Delta W_{ij}$ and $\Delta b_i$ are search directions, which are given by Equations (10) and (11) respectively, $\alpha$ is the learning rate.

$$\Delta W_{i+1,j}^{(l)} = -\frac{\partial}{\partial W_{i+1,j}^{(l)}} J(W, b) + \frac{\left\| \frac{\partial}{\partial W_{i+1,j}^{(l)}} J(W, b) \right\|^2}{\left\| \frac{\partial}{\partial W_{ij}^{(l)}} J(W, b) \right\|^2} \Delta W_{ij}^{(l)}, \tag{10}$$

$$\Delta b_{i+1}^{(l)} = -\frac{\partial}{\partial b_{i+1}^{(l)}} J(W, b) + \frac{\left\| \frac{\partial}{\partial b_{i+1}^{(l)}} J(W, b) \right\|^2}{\left\| \frac{\partial}{\partial b_i^{(l)}} J(W, b) \right\|^2} \Delta b_i^{(l)}, \tag{11}$$

where $\frac{\partial}{\partial W_{ij}^{(l)}} J(W, b)$ and $\frac{\partial}{\partial b_i^{(l)}} J(W, b)$ respectively represent the first-order partial derivatives of weights and offsets of the cost function $J(W, b)$. The update process of the weights and offsets of the feature layers (convolution layers and pooling layers) and the fully connected layers using the F–R algorithm is shown as follows.

| | Gradient update of F–R algorithm: |
|---|---|
| 1 | Completing the forward propagation process: initialize the weight W and offset b of each layer l to obtain the prediction result h. |
| 2 | Calculating the error between the predicted results and the real results by the cost function J. |
| 3 | Finding the first-order partial derivatives $J'_{Wi}$ and $J'_{bi}$ and calculate $J'_{Wi}$ and $J'_{bi}$ of each hidden layer l in reverse. |
| 4 | If the number of iterations is i = 1, the descending direction of each layer $\Delta W_i$ is the first-order partial derivative $J'_{Wi}$, and $\Delta b_i$ is $J'_{bi}$, and then go to step 6. |
| 5 | If the number of iterations i = 2, 3..., the descending direction of each layer $\Delta W_i$ is calculated according to Equation (10) and $\Delta b_i$ is calculated from Equation (11). |
| 6 | Substituting $\Delta W_i$ and $\Delta b_i$ into Equations (8) and (9), the weights $W_{ij}^{(l)}$ and offset $b_i^{(l)}$ of each layer are calculated, and the parameter update is completed. |
| 7 | Performing the next forward propagation until the cost function J converges or reaches the maximum number of iterations. |

Each search direction of the conjugate gradient algorithm is conjugate to each other, and these search directions are only a combination of the negative gradient direction and the search direction of the previous iteration. Therefore, the conjugate gradient algorithm only uses the first-order derivative information, which not only has the advantages of less storage and convenient calculation, but also overcomes the shortcoming of the slow convergence of the gradient descent algorithm, and becomes one of the most effective algorithms for solving large-scale nonlinear optimization. Therefore, in addition to the gradient descent method, the F–R algorithm has a small amount of computation of matrix operation in the optimization method and the absence of additional reference parameters. In addition, it will be demonstrated in the iteration time part of the experiment.

## 4. Experimental Results and Discussion

In this section, we experiment with the data presented in Section 2. To evaluate the performance of the proposed F–R CNN model, we set up comparison experiments with the traditional CNN model in changing the training samples number, disturbance magnitude, the batch training samples number and iteration time. In Section 4.4, we use the University of Pavia dataset to perform classification experiments and performance verification on the proposed method.

### 4.1. Dataset and Experimental Settings

#### 4.1.1. Dataset Description

The study area covers a variety of wetland landform types, and is now divided into five land cover types: Reed, tidal flats and bare land, mixture of Suaeda salsa and tamarisk, Spartina alterniflora and water. The land feature type marks (as shown in Figure 1b) are established based on the field data collected from the Yellow River Delta wetland field survey, high-resolution image data (as shown in Figure 1c) and the land cover interpretation marks (as shown in Table 2). The study area contains 262,144 sample points and 6090 of them were selected as training samples. The training samples selection follows the principle that the representative and the distribution of samples are as uniform as possible, and the relevant reference materials and expert interpretation knowledge are utilized. The distribution of training samples is shown in Figure 6. The number of training samples for each type of ground objects is shown in Table 4.

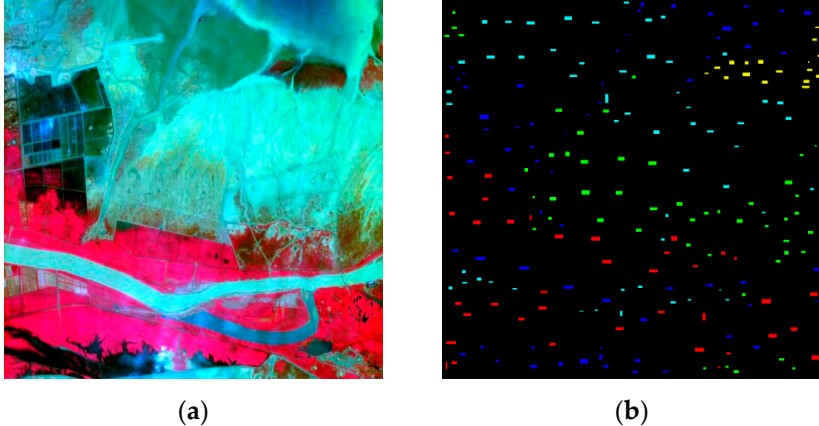

|      (a)      |      (b)      |
|---|---|

**Figure 6.** Training samples distribution. (**a**) False color image composite (band 15, 10 and 5); (**b**) ground truth of training samples.

#### 4.1.2. Experimental Settings

The F–R CNN structure includes two convolutional layers, two pooling layers, and two fully connected layers (Table 3). The network parameters are: Five convolution kernels of convolutional layer 1, eight convolution kernels of convolutional layer 2, and the two layers of pooling scale are one, the learning rate is 0.5, the number of batch training is two (except for Sections 4.2, 4.3.3 and 4.3.4), and the iteration is seven times. The percentage of correctly classified pixels is used as an indicator for accuracy evaluation.

**Table 4.** Land cover classes and the number of training samples for Yellow River Mouth wetland data set.

| Class | | | Samples | |
|---|---|---|---|---|
| No. | Color | Land Cover Types | Train | Test |
| 1 | | Reed | 1468 | 66121 |
| 2 | | Mixture of Suaeda-salsa and tamarisk | 1367 | 44811 |
| 3 | | Water | 1531 | 64274 |
| 4 | | Spartina alterniflora | 411 | 5926 |
| 5 | | Tidal flats and bare land | 1313 | 80942 |
| | | Total | 6090 | 262144 |

### 4.2. Experiment Results

The training samples of the study area and the expanded training samples were used to train the proposed F–R CNN model separately. The classification results will be compared with the traditional CNN model.

When using the original training samples (6090 samples) for model training, we set the batch training size to two samples and the number of iterations to seven. The classification accuracy of the F–R CNN model is 82.89%, which is 0.45% lower than the traditional CNN model. When using the expanded training samples (20 × 6090 samples) for model training, we set the disturbance range as [−0.02, 0.02]. The batch training size here is 10 samples and the number of iterations is seven. The classification accuracy of F–R CNN model is 84.93% which is 0.81% higher than the traditional CNN model. The above classification results are shown in Figure 7.

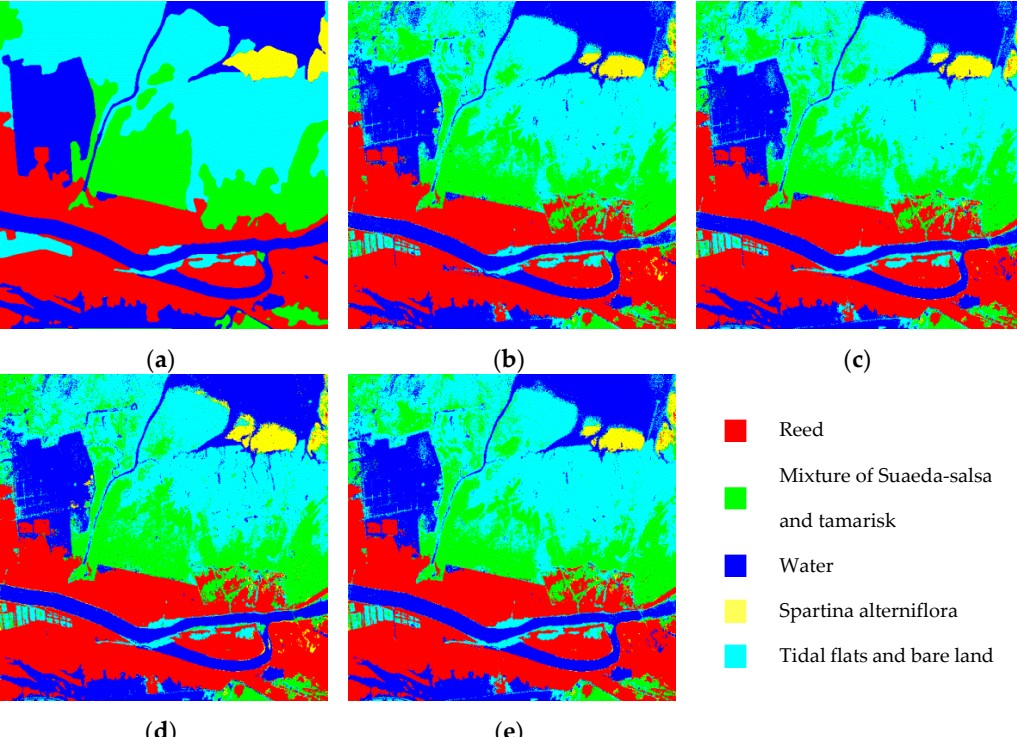

**Figure 7.** Classification results. (**a**) Ground truth image; (**b**) CNN with an accuracy of 83.34%; (**c**) F–R CNN with an accuracy of 82.89%. The training samples of (**b**) and (**c**) are the original training samples (6090 samples), and the batch training size is two samples. (**d**) CNN model with an accuracy of 84.02%; (**e**) F–R CNN model with an accuracy of 84.93%. The training samples of (**d**) and (**e**) are the expanded training samples (20 × 6090 samples), and the batch training size is 10 samples.

The model classification accuracy was improved by using extended samples for training. The accuracy of F–R CNN is increased by 2%, which indicates that the model has a certain anti-disturbance ability. After increasing the batch training size to 10 samples, per iteration time of F–R CNN is 1/5 of the two samples, which indicates that the model has a fast convergence advantage when dealing with a large number of samples.

*4.3. Discussion*

4.3.1. Increase Training Samples

The 6090 training samples selected in this paper only accounted for 2.3% of the samples in the study area. Due to the limited samples in the study area, it is considered to increase the number of training samples by increasing the disturbed samples. The basic idea is to make small disturbance to the training samples by adding random disturbance and form the new training samples with the original training samples. Now we use [−0.01, 0.01] as the disturbance range to generate different numbers of new samples. For example, a disturbance is added to the three times the training samples, and the generated new sample contains $4 \times 6090$ samples (6090 original training samples and $3 \times 6090$ disturbance samples).

The experiment generated a series of new samples, which were $4 \times n$ (n = 1, 2, 3, 4, 5) times the training samples. Multiple experiments were set for different new samples and the average accuracy was calculated. The number of batch training is two, and the iteration is seven times. The results are shown in the following table, where "1x" corresponds to the original training samples (6090 samples).

It can be seen from the classification results in Table 5 that, the accuracy of classification results appears as an irregular fluctuation, but most results are improved compared to the 1x training samples. The way of generating a new training set means increasing the number of training samples and containing a certain degree of noise. Therefore, the CNN model exhibit anti-disturbance ability under experimentally set noise interference, and so does the F–R CNN.

**Table 5.** Classification accuracy with increasing number of training samples.

| Training Samples | CNN(%) | Improved(%) | F-R CNN(%) | Improved(%) |
|:---:|:---:|:---:|:---:|:---:|
| 1x(original) | 83.34 | - | 82.89 | - |
| 4x | 82.96 | −0.38 | 83.16 | 0.27 |
| 8x | 83.54 | 0.20 | 84.15 | 1.26 |
| 12x | 84.56 | 1.22 | 83.67 | 0.78 |
| 16x | 82.61 | −0.73 | 82.79 | −0.10 |
| 20x | 83.47 | 0.13 | 82.74 | −0.15 |

4.3.2. Adjust the Disturbance Magnitude

In order to further explore the anti-disturbance performance of the proposed model, different degrees of disturbance are added to the training samples. Randomly disturbance samples were generated by adding random values of five different magnitudes (1%, 2%, 3%, 4%, and 5%) to training samples (corresponding to the disturbance ranges [−0.01, 0.01] to [−0.05, 0.05]), and together with the training samples respectively constitute new training samples, which is recorded as sample set A ~ E. Experiments were carried out with 4, 8, 12, 16, and 20 times the training samples. The number of batch training is two, and the iteration is seven times. In order to avoid the contingency of the disturbance, each disturbance set of different magnitudes was subjected to multiple experiments, and the mean value was used as the classification result, as shown in Table 6.

**Table 6.** Overall classification accuracy of different training sample sets. The value in parentheses is the increased value compared to the classification result of the original training sample set.

| Training Set | 4x (%) | | 8x (%) | | 12x (%) | | 16x (%) | | 20x (%) | |
|---|---|---|---|---|---|---|---|---|---|---|
| | CNN | FR-CNN | CNN | FR-CNN | CNN | FR-CNN | CNN | FR-CNN | CNN | FR-CNN |
| A (1%) | 82.96 (−0.38) | 83.16 (+0.27) | 83.54 (+0.2) | 84.15 (+1.26) | 84.56 (+1.22) | 83.67 (+0.78) | 82.61 (−0.73) | 82.79 (−0.10) | 83.47 (+0.13) | 82.74 (−0.15) |
| B (2%) | 83.67 (+0.33) | 83.77 (+0.88) | 84.98 (+1.64) | 84.46 (+1.57) | 84.05 (+0.71) | 83.34 (+0.45) | 83.49 (+0.15) | 82.31 (−0.58) | 84.28 (+0.94) | 83.39 (+0.50) |
| C (3%) | 83.53 (+0.19) | 83.13 (+0.24) | 84.62 (+1.28) | 84.52 (+1.63) | 84.44 (+1.10) | 82.69 (−0.20) | 83.87 (+0.53) | 82.88 (−0.01) | 84.12 (+0.78) | 82.75 (−0.14) |
| D (4%) | 83.55 (+0.21) | 82.54 (−0.35) | 84.73 (+1.39) | 83.85 (+0.96) | 83.99 (+0.65) | 83.47 (+0.58) | 83.31 (−0.03) | 83.19 (+0.30) | 83.90 (+0.56) | 82.29 (−0.60) |
| E (5%) | 82.82 (−0.52) | 82.58 (−0.31) | 84.68 (+1.34) | 84.07 (+1.18) | 84.24 (+0.90) | 83.22 (+0.33) | 83.11 (−0.23) | 82.84 (−0.05) | 82.96 (−0.38) | 82.76 (+0.13) |

It can be seen from Table 6 that most of the classification accuracies of both two models are improved with an increase in the number of samples. For the increase of different degrees of disturbance samples, the classification results of the models show different degrees of fluctuations. The variation of the magnitude of the disturbance will cause irregular fluctuations in the small range of the classification accuracy of the model. The improved accuracy of F–R CNN ranged from −0.6 to 1.63, and the traditional CNN ranged from −0.73 to 1.64. The above experimental results show that the F–R CNN model under the influence of small noise within a certain range has anti-interference ability that is not inferior to CNN, and still provides good classification results and stability.

The training time of the F–R CNN model and the traditional CNN model is statistically analyzed. The traditional CNN model has a little training time when the number of iteration steps is constant. Comparing the classification accuracy of the two models, it can be seen that although the traditional CNN model has less training time, its classification accuracy is lower. However, the F–R CNN classification accuracy is improved when the training time is not too much. The reason may be that the conjugate gradient can reach the optimal value faster under the fixed iterative steps, but the time consumption for finding the optimal value is more than that of the gradient descent algorithm. So F–R CNN converges faster with higher precision. (It will be discussed in the Section 4.3.4)

### 4.3.3. Change the Number of Batch Training Samples

When the number of new training samples increased by 20 times after the disturbance, the training time was also increased by 20 times. Therefore, we consider increasing the number of batch training samples. When other parameters of the network model are unchanged, we change the number of batch training samples to perform training and classification in different disturbed training samples. The iteration is seven times. The experimental results are shown in Table 7.

Referring to Table 7 and Figure 8, it can be seen that for different batch training sizes, when the number of batch training samples increases from 20 to 580, the overall classification accuracy of both two models decreases. The CNN model has a large decline. When the batch training size is 350, its accuracy has dropped below 77.82%, which is significantly lower than that of the F–R CNN model. However the F–R CNN is still above 80.72%, which is 2.9% higher than the traditional one. This is because the conjugate gradient algorithm has a faster convergence speed than the gradient descent algorithm under a limited number of iterations, and its storage amount in the calculation process is small, so it has an advantage in solving large nonlinear optimization problems.

Take the 2% disturbance level as an example, the classification results of the two models are shown in Figure 9. When the batch training size is 580, compared with the F–R CNN model, the CNN model is heavily confused with the mixture of Suaeda salsa and tamarisk, Tidal flats and bare land and water, as shown in Figure 9d.

**Table 7.** Overall classification accuracy of different batch training sizes in each disturbance interval. A, B, C, D and E represent 20 times the training samples of different disturbance levels.

| Batch Size | A (1%) (%) | | B (2%) (%) | | C (3%) (%) | | D (4%) (%) | | E (5%) (%) | |
|---|---|---|---|---|---|---|---|---|---|---|
| | CNN | FR-CNN | CNN | FR-CNN | CNN | FR-CNN | CNN | FR-CNN | CNN | FR-CNN |
| 20 | 84.14 | 83.65 | 84.33 | 83.88 | 84.02 | 83.97 | 84.13 | 83.97 | 84.16 | 84.14 |
| 40 | 83.19 | 84.04 | 83.18 | 84.27 | 83.14 | 83.96 | 83.30 | 84.02 | 83.00 | 84.05 |
| 60 | 82.35 | 83.92 | 82.28 | 83.96 | 82.33 | 83.84 | 82.49 | 83.89 | 82.19 | 83.84 |
| 75 | 82.13 | 83.50 | 82.07 | 83.47 | 82.08 | 83.46 | 82.21 | 83.47 | 82.00 | 83.39 |
| 100 | 81.74 | 82.52 | 81.70 | 82.48 | 81.70 | 82.53 | 81.84 | 82.66 | 81.63 | 82.34 |
| 120 | 81.53 | 82.22 | 81.50 | 82.07 | 81.49 | 82.19 | 81.58 | 82.32 | 81.44 | 82.03 |
| 140 | 81.32 | 82.04 | 81.29 | 81.95 | 81.27 | 81.99 | 81.39 | 82.16 | 81.24 | 81.90 |
| 175 | 80.87 | 82.07 | 80.85 | 82.06 | 80.83 | 82.03 | 80.91 | 82.18 | 80.80 | 81.91 |
| 203 | 80.35 | 81.80 | 80.34 | 81.78 | 80.31 | 81.72 | 80.39 | 81.89 | 80.30 | 81.65 |
| 232 | 79.74 | 81.50 | 79.73 | 81.45 | 79.70 | 81.45 | 79.76 | 81.58 | 79.69 | 81.36 |
| 280 | 78.92 | 81.31 | 78.90 | 81.28 | 78.88 | 81.26 | 78.94 | 81.38 | 78.87 | 81.24 |
| 300 | 78.56 | 81.20 | 78.56 | 81.18 | 78.53 | 81.16 | 78.58 | 81.26 | 78.52 | 81.11 |
| 350 | 77.82 | 80.78 | 77.80 | 80.77 | 77.77 | 80.74 | 77.82 | 80.80 | 77.76 | 80.72 |
| 420 | 76.78 | 79.99 | 76.76 | 79.97 | 76.76 | 79.93 | 76.77 | 80.01 | 76.75 | 79.93 |
| 580 | 73.60 | 78.64 | 73.57 | 78.62 | 73.60 | 78.60 | 73.60 | 78.66 | 73.60 | 78.60 |

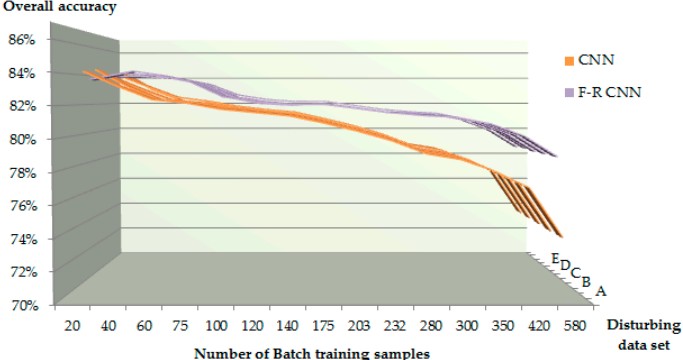

**Figure 8.** Classification accuracy evaluation of different batches of training numbers in two models. Data A, B, C, D and E represent 20 times the training samples of different disturbance levels.

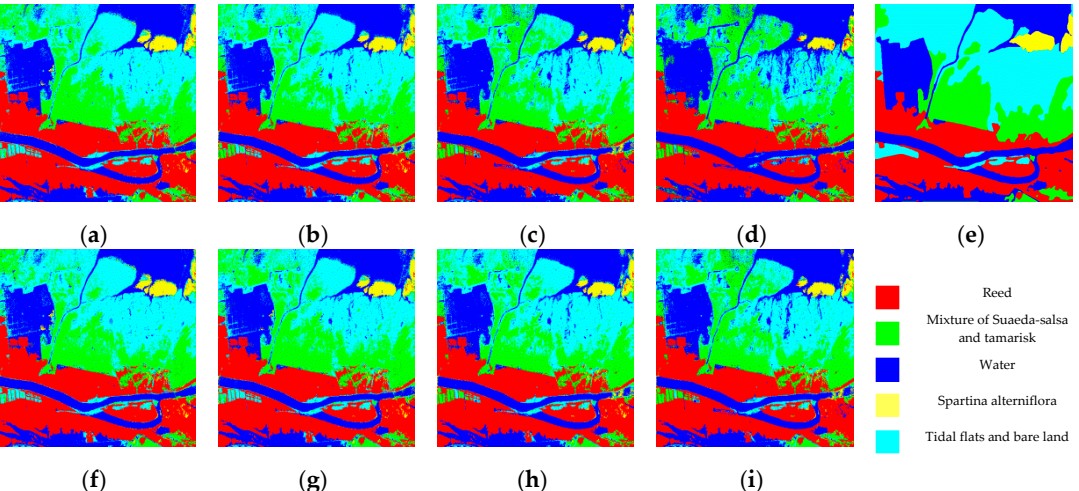

**Figure 9.** Classification results of 20x training sample with 2% disturbance level. (**a–d**) CNN model classification results (batch size = 40, 120, 300, and 580); (**e**) ground truth image, (**f–i**) F-R CNN model classification results (batch size = 40, 120, 300, and 580).

4.3.4. Iteration Time

Computation time consumption is inevitably generated while using a large number of training samples for network training. In the experiments of the previous subsection, it was found that when the batch training size increased, both of per iteration time and the overall training time reduced. However, with the number of batch training samples increasing, the classification accuracy of the model declined. This is because when the batch training size increases, the memory utilization is improved, and the parallelization efficiency of the matrix multiplication is improved; the number of iterations required for per iteration (total training sample) is reduced, and the processing speed for the same amount of data is accelerated.

The experiment was carried out in the 20 times the training samples with 2% disturbance level. The number of iterations is seven. The cost function convergence curves when training the classification model are shown in Figure 10. Comparing the two model curves, it can be seen that F–R CNN has better convergence under the same iteration steps. Especially when it is a large number of batch training samples, F–R CNN can converge more quickly, which makes the model training faster and gets the result better.

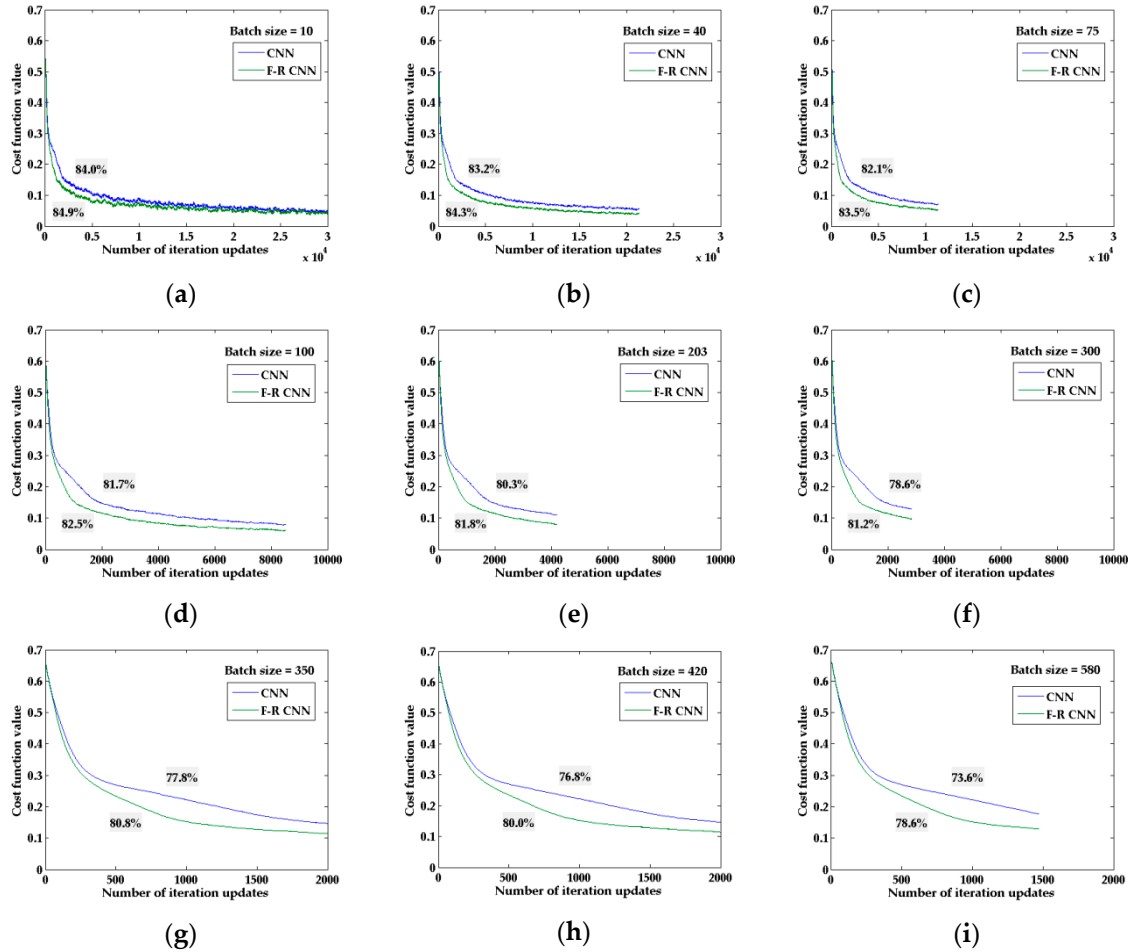

**Figure 10.** Cost function convergence curves. (**a**–**i**) Batch training sizes are 10, 40, 75, 100, 203, 300, 350, 420, and 580 respectively. The classification accuracy of the two models is marked at the corresponding convergence curve.

Taking the cost function less than 1% as the termination condition for the iteration, comparing the model iteration steps and calculation time of different batch training sizes, and the experimental results are shown in Tables 8 and 9. As the batch training size increases, on the one hand, the speed of processing the same amount of data per iteration increases, and on the other hand, the number of

iterations required to achieve the same accuracy increases. Therefore, when the batch training size increased to a certain value, the optimization in time and accuracy will be both achieved. The absolute time in the tables is per-unit processed.

**Table 8.** Convergence of traditional CNN models in different number of batch training samples.

| CNN Batch Size | 10 | 20 | 50 | 120 | 150 | 280 | 300 |
|---|---|---|---|---|---|---|---|
| Iteration step | 200 | 200 | 200 | 200 | 200 | 200 | 200 |
| Number of steps with error rate <0.01 | 2 | 10 | 31 | 139 | 168 | - | - |
| Time of error rate <0.01 | **1.0** | 2.7 | 4.0 | 9.5 | 11.0 | - | - |
| Average iteration time per step | 0.50 | 0.27 | 0.13 | 0.07 | 0.07 | - | - |

**Table 9.** Convergence of F–R CNN model in different number of batch training samples.

| F-R CNN Batch Size | 10 | 20 | 50 | 120 | 150 | 280 | 300 |
|---|---|---|---|---|---|---|---|
| Iteration step | 200 | 200 | 200 | 200 | 200 | 200 | 200 |
| Number of steps with error rate <0.01 | 2 | 7 | 26 | 43 | 43 | 167 | - |
| Time of error rate <0.01 | 1.3 | 2.4 | 4.0 | 4.0 | 3.6 | 11.0 | - |
| Average iteration time per step | 0.64 | 0.34 | 0.16 | 0.09 | 0.08 | 0.07 | - |

We can see from the table that for different batch training numbers, the time of per iteration of the two models is close, and the calculation amount of F–R CNN is slightly larger than the traditional one. However, when the number of batch training samples is increasing, the training time of F–R CNN is gradually approaching and eventually below the traditional model. When it gets to 150, the time consumption when the cost function error rate is less than 0.01 is 3.6 (the value is per-unit processed in above table), which is 67% shorter than the time taken by the CNN model. It highlights the advantage of the fast convergence of the proposed model. Comparing the convergence of the two models, we find that for different batch training sizes, the F–R CNN model has more time in per iteration to achieve the same precision, but the number of iteration steps required is small, and the final iteration time is significantly less than the traditional CNN model. This shows that the conjugate gradient algorithm converges faster during the training of the model. Therefore, in the specific model training, if the precision needs to be specified, the F–R CNN model takes less time; if the number of iterations is specified, it can achieve higher classification accuracy.

Combining the experimental results of the two models, it can be concluded that within a reasonable range, the more the number of batch training samples, the more accurate the direction of the decline, and the smaller the training shock. However, when the number of batch training exceeds a certain level, although the number of iterations required for per iteration (total training sample) is reduced, the number of iterations taken to achieve the same accuracy is greatly increased. Therefore, the correction of the parameters becomes slower, and the determined downward direction basically no longer changes. Therefore, by controlling the number of batch training within a reasonable range, the classification accuracy and consumption time can be effectively balanced.

## 4.4. Additional Application of the Proposed Method

In this section we apply the proposed model to the Pavia University dataset (shown in Figure 11), which is one of the public hyperspectral image benchmarks, and we will use it as an example to test our proposed classification methods. The Pavia University scene has $610 \times 340$ pixels and 103 spectral bands. The geometric resolution is 1.3 meters. There are nine types of ground truth samples in this image. The number of training samples and test samples for each class is shown in Table 10.

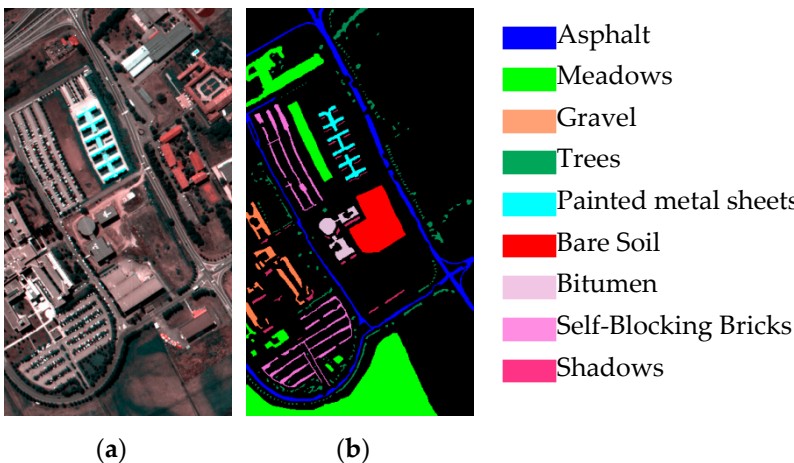

(**a**)  (**b**)

**Figure 11.** Pavia University dataset. (**a**) False color image composite (band 50, 27 and 17); (**b**) ground truth image.

**Table 10.** Land cover classes and the number of samples for Pavia University dataset.

| Class | | Samples | |
|---|---|---|---|
| No. | Land Cover Types | Train | Test |
| 1 | Asphalt | 332 | 6299 |
| 2 | Meadows | 933 | 17,716 |
| 3 | Gravel | 105 | 1994 |
| 4 | Trees | 154 | 2910 |
| 5 | Painted metal sheets | 68 | 1277 |
| 6 | Bare Soil | 252 | 4777 |
| 7 | Bitumen | 67 | 1263 |
| 8 | Self-Blocking Bricks | 185 | 3497 |
| 9 | Shadows | 48 | 899 |

We directly experiment with the Pavia University dataset using the same network structure, repeating the structure and parameters as follows. It includes two convolutional layers, two pooling layers and two fully connected layers, and five convolution kernels of convolutional layer 1, eight convolution kernels of convolutional layer 2, and the two layers of pooling scale are 1, the learning rate is 0.5, and the iteration is 20 times. The percentage of correctly classified pixels of test samples is used as an indicator for accuracy evaluation. In Figure 12, we compare the F–R CNN model with the traditional CNN, and give the classification results of the original sample and the sample set after the sample expansion (disturbance level 4%, 20x samples) and batch training number tuning.

We tested the F–R CNN classification model in three aspects: Anti-disturbance ability, batch computing adaptability and convergence speed. The expanded datasets are 20x training samples containing the disturbance. Table 11 shows the classification accuracy of 20x training samples with different perturbations (1%–5%) and the batch training size is two samples. It can be seen that the test accuracy of the two models was improved after expanding the training samples, especially the traditional one. Both models can maintain anti-interference.

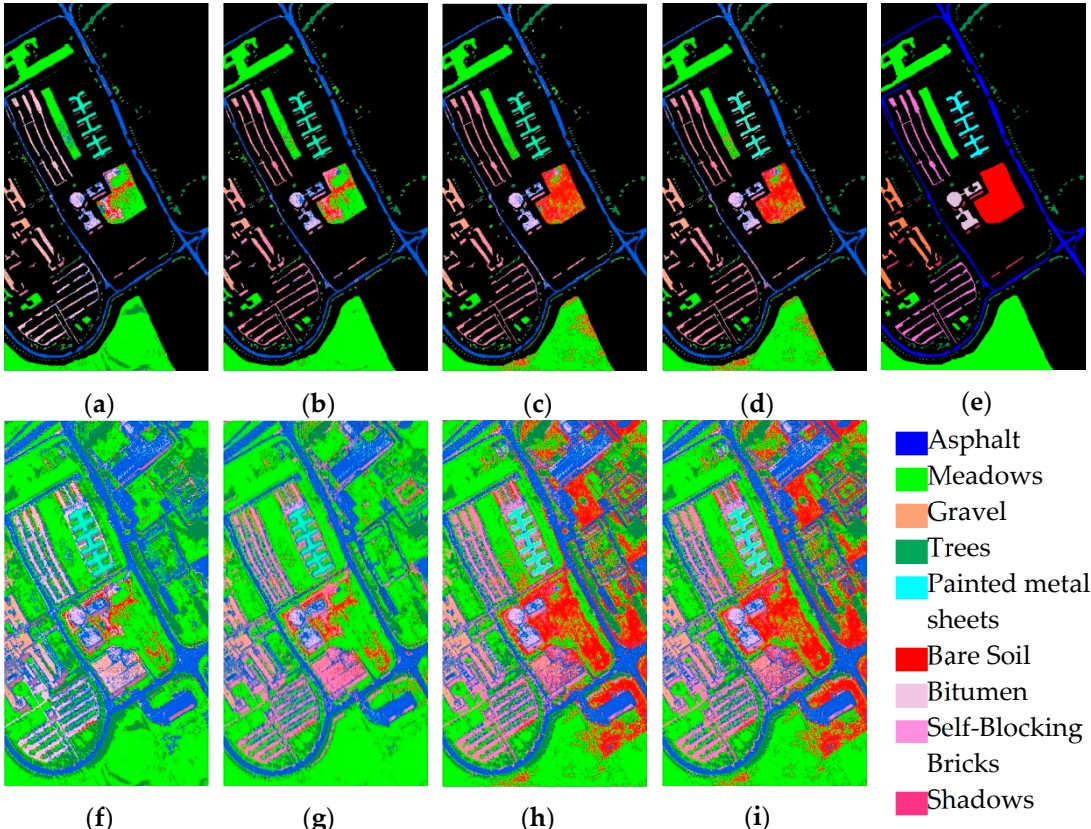

**Figure 12.** Classification results. (**a**) CNN with an accuracy of 80.14%; (**b**) F–R CNN with an accuracy of 81.26%. The training samples of (**a**) and (**b**) are the original training samples (2144 samples), and the batch training size is two samples. Image (**f**) and (**g**) are the global classification result s corresponding to (**a**) and (**b**) respectively; (**c**) CNN with an accuracy of 85.21%; (**d**) F–R CNN with an accuracy of 86.93%. The training samples of (**c**) and (**d**) are the expanded training samples (20 × 2144 samples), and the batch training size is eight samples. Image (**h**) and (**i**) are the global classification results corresponding to (**c**) and (**d**) respectively; (**e**) ground-truth image.

**Table 11.** Test accuracy of different training sample sets.

| Method | CNN (%) | | F-R CNN (%) | |
|---|---|---|---|---|
| Training Set | Test Accuracy | Improved | Test Accuracy | Improved |
| 1X | 80.14 | - | 81.26 | - |
| 20X-A | 84.96 | 4.82 | 83.08 | 1.82 |
| 20X-B | 85.15 | 5.01 | 84.07 | 2.81 |
| 20X-C | 85.55 | 5.41 | 84.08 | 2.82 |
| 20X-D | 84.95 | 4.81 | 83.34 | 2.08 |
| 20X-E | 84.98 | 4.84 | 83.16 | 1.9 |

The batch computing adaptability can be visually seen through Figure 13a. When the number of batch training samples increased from 8 to 320, the classification accuracy of both models showed a downward trend. The accuracy of CNN is significantly reduced than that of the proposed model, which indicates that the proposed model can guarantee a high classification accuracy and stability during large-scale sample training. Figure 13b shows the convergence curve of the two models during training. When the number of iterations is constant, the cost function convergence curve of the proposed model is lower than that of the traditional model for different batch training numbers, and the classification results of the proposed model is finally higher than the traditional one (according to Figure 13a).

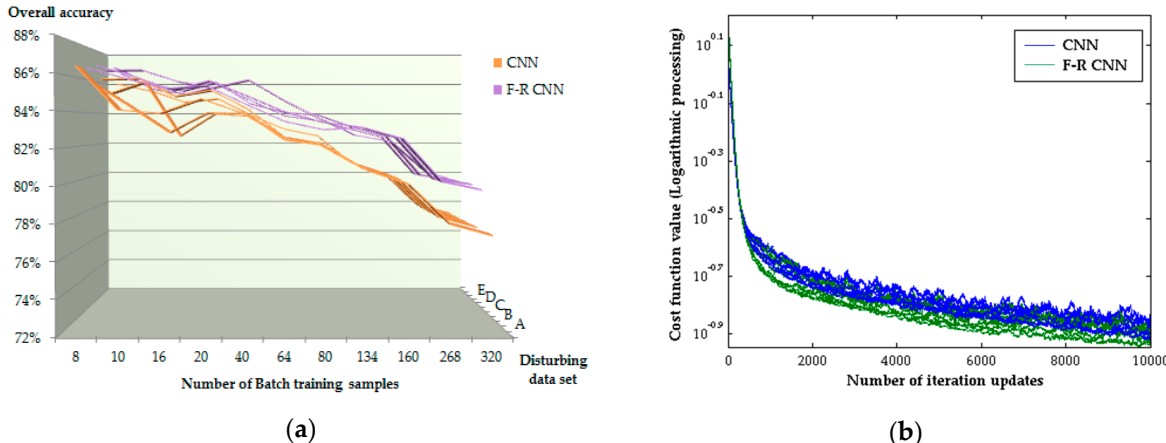

**Figure 13.** Test accuracy curve and cost function convergence curve. (**a**) Test accuracy evaluation of different batch training numbers in two models. Data A, B, C, D and E represent 20 times the training samples of different disturbance levels; (**b**) cost function convergence curves.

Through the above analysis of the classification experiment results in Pavia University dataset, we can conclude that the proposed model can effectively classify it. Additionally, the rules and conclusions obtained by the proposed method in the application of the Pavia University dataset are consistent with that of the Yellow River estuary wetland dataset in aspects of anti-disturbance ability, batch computing adaptability and convergence speed.

## 5. Conclusions

In this paper, we proposed a CNN model based on Fletcher–Reeves algorithm (F–R CNN), which uses the Fletcher–Reeves algorithm for gradient updating to optimize the model convergence performance in classification. To solve the problem of fewer optional training samples in practical applications, we further proposed a method of increasing the number of training samples by adding spectral perturbations and tested the anti-interference ability of the proposed method. In this experiment, we comprehensively evaluated the proposed model based on the classification of CHRIS hyperspectral imagery covering coastal wetlands. We analyzed the anti-interference and convergence performance of the proposed model in terms of different training datasets, different batch training sample numbers and iteration time. After increasing the number of training samples and adjusting the number of batch training samples, the classification results of the two models is effectively improved, especially the accuracy of the proposed model is improved by 2% and is higher than that of the traditional one. Compared with the traditional CNN model, the experimental results show that the proposed model has the characteristics of the anti-disturbance ability, batch computing adaptability and convergence speed. Finally, we used the Pavia University dataset to evaluate the proposed model, and its results are consistent with the above statement of the CHRIS dataset. The proposed model provides a new idea for the optimization of the CNN model, and it can face the challenge of large computation while improving the model convergence ability.

**Author Contributions:** M.Y. provided the conceptualization, developed the methodology and revised the manuscript. C.C. performed the experiments and analysis, and wrote the manuscript. M.Y. and R.G.B. provided financial support.

**Funding:** This research was funded by National Natural Science Foundation of China (61890964, 41206172, and 41706209).

**Acknowledgments:** We gratefully acknowledge the European Space Agency (ESA) on the new generation of microsatellite Project for On-Board Autonomy (PROBA) for providing the CHRIS productions.

**Conflicts of Interest:** The authors declare no conflict of interest.

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
