# Peer review of "A Convolutional Neural Network with Fletcher–Reeves Algorithm for Hyperspectral Image Classification"

_remotesensing, doi:10.3390/rs11111325_

Round 1

Reviewer 1 Report

This paper presents a CNN model based on Fletcher-Reeves algorithm (F-R CNN) to replace the gradient descent algorithm in the traditional model in order to solve the problem of slow convergence of CNN model based on gradient descent. My comments are detailed below:

1.  The abstract requires significant modification. The abstract should briefly present: intro, problem, method, results and finally conclusions; you have to highlight the scientific problem of this research and please give a brief description of the used algorithm.

2.   Line 18: you jumped to the conclusions before presenting the results, please move this sentence after the results.

3.       Line 19: you stating different scenarios without stating the original one.

4.       Line 23: it is a kind of repetition of Line 18.

5.       Line 18: remove the colon.

6.       Line 70: two sets of experiments >> two experiments

7.       Line 79: what is CHRIS? Please define in full here instead of Line 83.

8.       Line 84: ESA, please define

9.       Line 91: any justifications on why using this mode?  Description/complexity

10.   Line 93: HDF, please define

11.   Line 104: It is not clear, how did you generate the reference data in figure 1b?

12.   Figure 2: separate these two figures into two different frames

13.   Line 318: how did you come with this conclusion? According to table 5, the improvement is unstable with increasing training samples.  Also, the accuracy of F-R CNN is not always higher than CNN (e.g., 1x, 12x, 20x training samples? Please clarify

14.   Line 337: decreases >> is

15.   Line 338-339: “F-R CNN has stronger and higher” I see traditional CNN has also anti-interference ability and can provide high accuracy. Please clarify

16.   Line 393: in the following table >> in Table 7 and 8

17.   Line 397: in the table was per-unit processed >> in the tables is per-unit processed

18.   Support the conclusions by results

19.   Line 421: “So it has the better…” it is very generic sentence, please remove it

20.   Line 523: alignment

Author Response

Response to Reviewer 1 Comments

Thank you for your comments, we responded to each of comments and revised the paper.

Point 1: The abstract requires significant modification. The abstract should briefly present: intro, problem, method, results and finally conclusions; you have to highlight the scientific problem of this research and please give a brief description of the used algorithm.

Response 1: The abstract was significantly modified according to the comments. The contents of the abstract are supplemented, and the main work and methods of this paper are highlighted.

Point 2: Line 18: you jumped to the conclusions before presenting the results, please move this sentence after the results.

Response 2: It was adjusted according to the comments.

Point 3: Line 19: you stating different scenarios without stating the original one.

Response 3: Modifications was added to the abstract.

Point 4: Line 23: it is a kind of repetition of Line 18.

Response 4: It was adjusted according to the comments.

Point 5: Line 18: remove the colon.

Response 5: It was modified in the abstract.

Point 6: Line 70: two sets of experiments >> two experiments

Response 6: It was modified accordingly.

Point 7: Line 79: what is CHRIS? Please define in full here instead of Line 83.

Response 7: It was defined in Line 92.

Point 8: Line 84: ESA, please define

Response 8: It was defined in Line 98.

Point 9: Line 91: any justifications on why using this mode?  Description/complexity

Response 9: It was added in line 107-109.

The CHRIS image in Mode 2 has the widest spectral range of 406~1035 nm.0° is a top view imaging, which has the best image quality among the five imaging angles, while other angles have strong surface reflection information, which is not suitable for object classification. In addition, the top view also conforms to the imaging geometry of hyperspectral remote sensing images. Therefore, this paper uses the 0° image in mode 2 to carry out the experiment.

Point 10: Line 93: HDF, please define

Response 10: It was added in line 109.

Point 11: Line 104: It is not clear, how did you generate the reference data in figure 1b?

Response 11: It was added in line 122-127.

Our team has gone to the Yellow River Mouth Wetland for field reconnaissance (including field photo and field spectral acquisition of the survey site) every year since 2000, and we are very familiar with the types of land cover and their distribution at the survey route and collection point. Therefore, we have rich experience in the identification of the type of ground cover in the Yellow River Mouth Wetland area. The production of the feature map of the study area is based on the above experience, and its production process is:

Referring to the terrain interpretation table (Tab.2), the vector map was obtained by information extraction of all terrain types of images in the research area (Fig.1a). Then, we made a detailed correction with reference to the high spatial resolution image and finally generated Figure 1b.

Point 12: Figure 2: separate these two figures into two different frames

Response 12: It was adjusted according to the comments in Figure 2.

Point 13: Line 318: how did you come with this conclusion? According to table 5, the improvement is unstable with increasing training samples.  Also, the accuracy of F-R CNN is not always higher than CNN (e.g., 1x, 12x, 20x training samples?) Please clarify.

Response 13: It was modified on Line 353-357.

The conclusions here were revised. It can be seen from Table 5 that since the classification results of the original samples (1x samples) of the two models have a certain difference, the relative improvement of the accuracy of the same model is compared here. Both models have good anti-interference ability, so it can be preliminarily determined that F-R CNN has anti-interference performance. We have revised the conclusions here and made a more objective statement.

Point 14: Line 337: decreases >> is

Response 14: It was adjusted according to the comments.

Point 15: Line 338-339: “F-R CNN has stronger and higher” I see traditional CNN has also anti-interference ability and can provide high accuracy. Please clarify.

Response 15: It was modified on Line 373-378.

The conclusions were revised. The results in Table 6 show that both models have anti-interference ability and can provide effective classification. Therefore, the conclusion is that F-R CNN can guarantee certain anti-interference, which is the same as CNN.

Point 16: Line 393: in the following table >> in Table 7 and 8

Response 16: It was adjusted according to the comments on Line 433.

Point 17: Line 397: in the table was per-unit processed >> in the tables is per-unit processed

Response 17: It was adjusted according to the comments on Line 437.

Point 18: Support the conclusions by results

Response 18: It was modified in conclusions according to the experimental results.

Point 19: Line 421: “So it has the better…” it is very generic sentence, please remove it

Response 2: It was removed according to the comments.

Point 20: Line 523: alignment

Response 20: It was adjusted according to the comments on Line 646.

Reviewer 2 Report

This paper proposed a CNN model based on Fletcher-Reeves algorithm (F-R CNN) for hyperspectral image classification. To this end, F-R CNN employed Fletcher-Reeves algorithm to replace the gradient descent algorithm in the traditional model and optimized the network parameters to classify HSIs. Comprehensive experiments demonstrated the effectiveness of the proposed method.

Generally speaking, the innovation of this paper is OK. The reviewer recommends this paper for publication after some revision.

Some specific comments are as follows:

1. In section 3.1, how to normalize the gray values of the HSI sample data? The normalization method used in this paper should be clarified.

2. One of the advantages of the proposed method is less storage, so it is better to add the analysis of storages for different methods.

3. What are the numbers of the test samples for different land cover types?

4. The experimental dataset used in this paper is CHRIS hyperspectral remote sensing data which covers coastal wetlands. As the reviewer knows, there are some public HSI benchmarks (such as University of Pavia, Salinas, and Indian Pines) which are always used to test the HSI classification methods. How about the effectiveness of the proposed method on these datasets? If applicable, the authors should add the experimental results on these datasets.

5. Some detailed parameters should be provided, including x, y, the learning rate lambda_k, and so on.

6. What is the number of iterations for the network training of the comparison experiments in Section 4.3.1, 4.3.2, and 4.3.3?

7. In Figure 10, the experiments only give the comparisons of cost function values with the number of iteration increasing. How about the comparisons of classification accuracies with the number of iteration increasing?

8. Some relevant works concerning hyperspectral image classification should also be cited, such as "Exploring hierarchical convolutional features for hyperspectral image classification" and "Learning Compact and Discriminative Stacked Autoencoder for Hyperspectral Image Classification".

Author Response

Response to Reviewer 2 Comments

Thank you for your comments, we responded to each of comments and revised the paper.

Point 1: In section 3.1, how to normalize the gray values of the HSI sample data? The normalization method used in this paper should be clarified.

Response 1: It was added in Line 170-175.

Processing matrices by mapping row minimum and maximum values to [-1 1]. Each row here represents the spectral vector of each sample. The algorithm formula is as follow.

y = ( y_max - y_min ) * ( x - x_min ) / ( x_max - x_min ) + y_min,

Where y represents the normalized vector, y_max and y_min respectively represent normalized thresholds 1 and -1, x is the sample spectral vector, and x_max and x_min are the maximum and minimum values of the sample spectral vector.

Point 2: One of the advantages of the proposed method is less storage, so it is better to add the analysis of storages for different methods.

Response 2: It was analysed in line 230-234, 247-249, 281-284, 440-446.

The less storage of the proposed method is compared to other optimization methods such as Newton’s method. The estimate of the calculated storage amount can be reflected in the amount of calculation. F-R algorithm performs a derivative operation in the calculation process, which is the same as the gradient descent method. It only adds a small amount of computation when calculating the conjugate direction. Although the Newton’s method converges quickly, it needs to calculate the second derivative and the matrix inversion, which is a large amount of computation and will take up a lot of computational memory. Therefore, the F-R algorithm has the advantages of simple calculation, small operation memory and fast convergent.

Point 3: What are the numbers of the test samples for different land cover types?

Response 3: The number of samples for each type of global verification image was supplemented in Table 4.

In order to avoid the contingency caused by the artificial selection of test samples, this paper conducts global verification on the study area, which makes the classification results more objective and more rigorous.

Point 4: The experimental dataset used in this paper is CHRIS hyperspectral remote sensing data which covers coastal wetlands. As the reviewer knows, there are some public HSI benchmarks (such as University of Pavia, Salinas, and Indian Pines) which are always used to test the HSI classification methods. How about the effectiveness of the proposed method on these datasets? If applicable, the authors should add the experimental results on these datasets.

Response 4: Based on your suggestions, we added a new section 4.4 to validate the proposed method using the University of Pavia dataset and compare it to the traditional method.

Point 5: Some detailed parameters should be provided, including x, y, the learning rate lambda_k, and so on.

Response 5:  Parameters such as x, y, and learning rate are supplemented in Line 182-183, 216-219, 274.

Point 6: What is the number of iterations for the network training of the comparison experiments in Section 4.3.1, 4.3.2, and 4.3.3?

Response 6: The number of iterations was explained in Line 313 of section 4.1.2, and we have also supplemented section 4.3.1, 4.3.2, and 4.3.3 according to the modification opinions in Line 350, 365, 392.

Point 7: In Figure 10, the experiments only give the comparisons of cost function values with the number of iteration increasing. How about the comparisons of classification accuracies with the number of iteration increasing?

Response 7: According to the review opinions, the classification accuracy of each convergence curve of the two models was added in figures.

Figure 10 shows the convergence of the cost function of the two models during training. It can be seen that the cost function of the F-R CNN converges quickly and with high precision, while the CNN converges slowly with low classification accuracy. Regarding the influence of the number of iterations on the training results, we can know from Tables 7 and 8 that the convergence performance of the model can be more directly reflected by comparing the number of iterations required to achieve a certain precision.

Point 8: Some relevant works concerning hyperspectral image classification should also be cited, such as "Exploring hierarchical convolutional features for hyperspectral image classification" and "Learning Compact and Discriminative Stacked Autoencoder for Hyperspectral Image Classification".

Response 8: It was cited some relevant works concerning hyperspectral image classification in section1 according to your modification opinions.

Round 2

Reviewer 1 Report

No further comments.

Reviewer 2 Report

My comments have been addressed.